# TOWARDS COHERENT AND CONSISTENT
# USE OF ENTITIES IN NARRATIVE GENERATION

## ABSTRACT

Large pre-trained language models (LMs) have demonstrated impressive capabilities in generating long, fluent text; however, there is little to no analysis on their ability to maintain entity coherence and consistency. In this work, we focus on the end task of narrative generation and systematically analyse the long-range entity coherence and consistency in generated stories. First, we propose a set of automatic metrics for measuring model performance in terms of entity usage. Given these metrics, we quantify the limitations of current LMs. Next, we propose augmenting a pre-trained LM with a dynamic entity memory in an end-to-end manner by using an auxiliary entity-related loss for guiding the reads and writes to the memory. We demonstrate that the dynamic entity memory increases entity coherence according to both automatic and human judgment and helps preserving entity-related information especially in settings with a limited context window. Finally, we also validate that our automatic metrics are correlated with human ratings and serve as a good indicator of the quality of generated stories.

## 1 INTRODUCTION

Large pre-trained language models (LMs) (such as GPT-2 (Radford et al., 2019), GPT-3 (Brown et al., 2020), and models based on the Transformer-XL architecture (Dai et al., 2019)) have radically improved text generation, producing seemingly fluent text – Clark et al. (2021) even showed that non-expert human judges cannot distinguish between machine-written and human-authored texts, based on surface cues. Assuming the quality of generated text as given, most recent efforts have then focused on trying to control generation with a desired topic, factual information, or specific style (Keskar et al., 2019; Dathathri et al., 2019; Shin et al., 2020; Li & Liang, 2021). However, anecdotally, there are still common failure cases of machine generated text in terms of entity coherence and consistency, which are fundamental properties of language.

In this work, we specifically focus on the task of narrative generation in order to analyse and improve entity coherence and consistency. Entities play a central role in narratives, since they guide the plot, and all events revolve around them (Fludernik, 2002; Jannidis, 2009; Frow, 2014; Bamman et al., 2013). Despite the importance of entities, recent work has mainly emphasised on controlling the topic of the generated stories using outlines, keywords or other relevant knowledge (Xu et al., 2020; Rashkin et al., 2020; Fan et al., 2019; Goldfarb-Tarrant et al., 2020). At the same time, entity-related structure in narrative generation has been largely understudied for large-scale pre-trained LMs.

First, we propose a set of metrics for automatically measuring entity coherence and consistency. Based on these metrics, we observe that the current LMs fail to follow the patterns of entity usage we find in human-written narratives. Overall, the generated stories present significantly lower coherence and consistency, and this is especially evident for stories with complex events and many named entities. We further validate these observations by performing a human evaluation study, showing that our automatic metrics correlate with human judgment of entity coherence.

Next, in order to improve these properties in narrative generation, we propose augmenting a pre-trained LM with a dynamic entity memory. Motivated by prior work on language modeling (Clark et al., 2018; Ji et al., 2017), which uses dynamic entity representations for improving generation on smaller RNN-based models, we augment the LM with an entity memory and cross-attention blocks at each layer of the model for attending to entities that participate in the narrative.

(a) Examples of constructed entity prompts for WikiPlots and WritingPrompts. Notice the different types of entities included in the two datasets.

(b) Examples of entity-related issues in generated text. These examples have been extracted from the GPT-2 model when providing short prompts.

Figure 1: Task formulation: Entity-driven generation for increased coherence and consistency.

In contrast with prior work, we introduce an end-to-end trainable network with soft attention for performing reads and writes to the memory instead of separately training models to predict entity detection and reference. We also relax the hard constraints of Clark et al. (2018) and Ji et al. (2017), who only condition on one entity per step and update an entity representation only when encountering one of its mentions. Instead, in order to increase both efficiency in the context of transformer-based networks and flexibility of the entity-token mapping, we instead perform soft reads from the entity memory based on a cross-attention mechanism. Thus, our model can condition on multiple relevant entities, and update all slots depending on the cross-attention scores after regular intervals within the narrative. Moreover, we exploit token-level entity annotations in order to regularize the cross-attention scores and better guide the reads and writes to the entity memory.

We perform experiments on two narrative datasets, WritingPrompts (Fan et al., 2018) and WikiPlots,[1] and find that utilizing an entity memory especially increases entity coherence according to both automatic metrics and human judges. Moreover, we experiment with different scenarios, where the LM has access to a limited narrative context (i.e., varying smaller context windows), in order to simulate model behavior in settings with much longer narratives, such as books or screenplays. Since narratives of this length cannot fit into the LM's short-term memory, we investigate the loss of entity-related information as we move to later narrative sections. By measuring perplexity and uncertainty on entity mentions on the original stories, we find that the dynamic entity memory is able to preserve significantly more entity-related information in limited context settings.

## 2 TASK FORMULATION

This work aims at the exploration of entity coherence and consistency in the context of narrative generation. Entities play a central role in narratives and are crucial for the development and quality of the story (Jannidis, 2009; Frow, 2014; Bamman et al., 2013). According to Fludernik (2002), there can even be narratives without plot, but not without a human experiencer in their center. Narrative theories have also studied character archetypes with specific attributes and actions as a means for analysing them (Fludernik, 2002; Jung, 2014).

We formulate the task of entity-driven generation as conditional text generation on a set of given entities. Specifically, we identify and provide the gold entities that participate in a narrative via an entity prompt. Each entity may consist of more than one token and different entities are separated with a special separator token. Examples of entity prompts are presented in Figure 1a and details about their construction are given in Section 4.3. Our objective is to investigate the patterns of entity usage in generated stories in comparison with human-written ones.

More formally, we consider a LM that is conditioned on an entity prompt $\mathcal{P}$ and learns the distribution $p(x|\mathcal{P})$ for generating narratives. The LM is trained on sequences of raw narrative text prepended with the corresponding entity prompts. The LM operates autoregressively; that is, given $\mathcal{P}$ and the context generated so far $x_{\leq t} = \{x_0, x_1, ..., x_t\}$, the LM computes a distribution for the next word in the narrative. Next, we define metrics for automatically measuring entity coherence and consistency in both human-written and generated stories. We evaluate the proposed metrics against human ratings in Section 5.3.

---

[1]https://github.com/markriedl/WikiPlots

**Entity coherence**   Various local entity coherence metrics have been suggested in literature, such as distance-based clustering and linkage coefficients (Lioma et al., 2016) and local entity coherence (Barzilay & Lapata, 2008; Mesgar & Strube, 2014; Guinaudeau & Strube, 2013). However, current LMs present high local coherence when compared with human-written stories, giving the impression that coherence has been achieved. In contrast, during preliminary analysis of longer narratives, we observed that LMs still struggle with maintaining *long-range entity coherence* (see Figure 1b for a short incoherent example and Tables 7, 8, and 9 of the Appendix for longer examples of real generated text). Our main observation from generated stories is that LMs tend to drop the initial protagonists after a while and instead introduce new, irrelevant entities (details in Section 5). For quantifying this observation, we propose a new metric. We consider the protagonists of the narrative (i.e. the entities with the most mentions) and divide the narrative into $L$ equal sections. Next, we compute the maximum span of mentions for each protagonist $i$ as the maximum interval of sections where $i$ appears in: $C_i = s_{l_i} - s_{f_i}$. Here, $s_{f_i}$ and $s_{l_i}$ are the indices of the sections containing the first and last mentions respectively of entity $i$.

**Entity consistency**   Another important aspect that we evaluate in the context of entity usage is the attributes that are given to each entity throughout the narrative (see Figure 1b for an inconsistent example). Traditionally, narratives use archetypes for the protagonists (e.g., the "hero" and the "trickster"; Fludernik 2002; Jung 2014) with rich and diverse features, personalities and consistent actions. As a measure of how well-developed and consistent each entity is within the narrative, we measure attribute consistency. Specifically, given all mentions per entity in a story, we consider as the attributes of the entity all *verbs* and *adjectives* that appear in the same sentence as each of its mentions. Next, we compute the percentage of unique attributes $\mathcal{U}_i$ for the $i^{th}$ entity as follows:

$$\mathcal{U}_i = \frac{|\bigcup_{j=1,i\in E_j}^N \mathcal{A}_j| - |\bigcup_{j=1,i\in E_j}^N \mathcal{A}_j \bigcap \bigcup_{j=1,i\notin E_j}^N \mathcal{A}_j|}{|\bigcup_{j=1,i\in E_j}^N \mathcal{A}_j|} \tag{1}$$

where $N$ is the number of sentences in the story, $E_j$ are the entities that are mentioned in the $j^{th}$ sentence, $\mathcal{A}_j$ is the set of all attributes that appear in the $j^{th}$ sentence, and $|\cdot|$ is the size of the set.

## 3   METHOD

Our base LM is a pre-trained Transformer-XL (T-XL) model (Dai et al., 2019) conditioned on $\mathcal{P}$. The T-XL LM allows us to consider an extended context window within the narrative when computing token representations in self-attention by using a cache memory, where all intermediate representations of the $M$ tokens prior to the current context are stored and used for as context. In this work, we propose augmenting the pre-trained base LM with an entity memory (MNEMELM). For attending to the entity memory, we add new, randomly initialized cross-attention blocks in parallel with self-attention per layer resembling the architecture of adapters[2] (Houlsby et al., 2019). We propose using the entity memory together with the prompt for richer entity representations and to better preserve entity-related information over a long time horizon. This addresses two limitations of prompts:

1. They do not allow for more meaningful entity representations. For example, given a named entity such as "Sarah King", the tokens from the prompt do not provide any information related to who Sarah King is, or which the attributes of Sarah King are within the context of the narrative. In contrast, our dynamic entity memory can store attributes of the entity as they appear in the text, which offers more information beyond the surface form.

2. LMs eventually forget about the prompt when given long enough narratives (i.e. the prompt will fall out of the short-term memory of the LM). In contrast, our method can efficiently store entity-related information in a fixed-size memory and independently of the current context window. We demonstrate this empirically in Section 5.1.

**Memory initialization**   We first initialize the entity memory based on the information given in the prompt $\mathcal{P}$. Specifically, each memory slot $M_j, j \in [1, Z]$ represents one of the $Z - 1$ entities that participate in the narrative or corresponds to non-entity information (i.e. the $Z^{th}$ slot is reserved for entity-irrelevant information). Each of the entity-related slots is initialized based on the prompt

---

[2]In contrast to adapters, we find that just training the new parameters is insufficient for narrative generation.

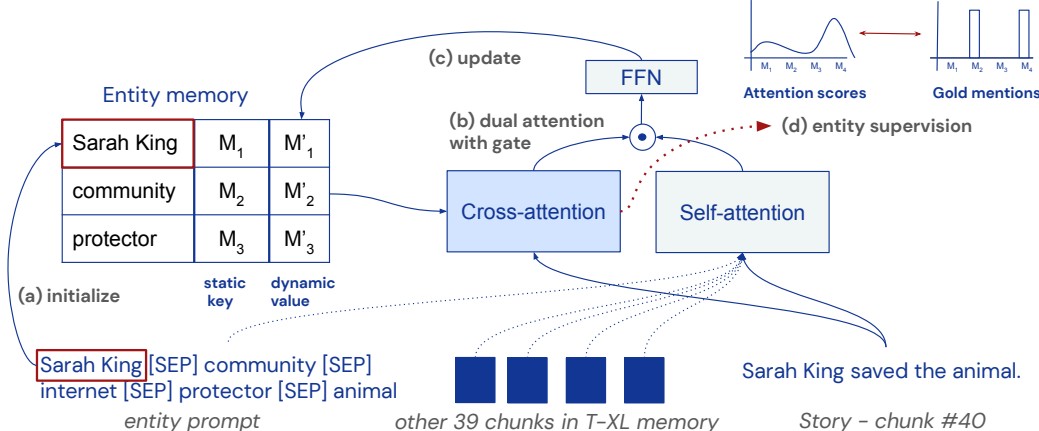

Figure 2: D-MNEMELM: The entity memory is initialized based on the (contextualised) embedding representations of the prompt tokens (a). Next, the narrative is processed chunk-by-chunk. At each model layer, there is a (pre-trained) self-attention block that considers all previous context, and a new, randomly initialized cross-attention block for attending to the entity memory. The two components are combined via a gating mechanism (b). Finally, the representation of the current chunk is used for updating the dynamic values of the entity memory (c). The cross-attention scores are regularized during training based on gold token-level entity mentions (d).

tokens that correspond to this entity (i.e. tokens allocated within two special separator tokens). For contextualizing the entity tokens before the memory initialization, we process the prompt via the LM and consider the output token-level embeddings. Next, the final representation for the $j^{th}$ slot is: $M_j = \frac{1}{K}\sum_{k=1}^{K} y_k$, where $K$ is the number of tokens that correspond to the $j^{th}$ entity and $y_k$ is the output embedding of the $k^{th}$ token.

**Conditioning on a dynamic entity memory (D-MNEMELM)** Each slot $M_j = [K_j, V_j]$ of the entity memory contains a static key $K_j$ (i.e. a fixed surface entity representation) and a dynamic value $V_j$ (i.e. a frequently updated representation based on narrative context), initialised as described above. To update the memory, we divide the narrative into equal-length chunks, update the entity memory after processing each chunk, and use the T-XL memory to store the previous chunks.

At each layer of the pre-trained LM, we add a new, randomly initialized cross-attention block that operates in parallel with the pre-trained self-attention one. The cross-attention block takes as input the representation $x_i$ of the $i^{th}$ token (either an embedding or intermediate representation) and all memory slots $M = [K, V]$, and computes an entity-aware representation $e_i$ as follows:

$$a_{it} = \text{softmax}\left(\frac{W_Q^t x_i W_K^t M^T}{\sqrt{d_M}}\right), t \in [1, H] \tag{2}$$

$$M_{it}^{att} = W_M^t a_{it} M \quad e_i = W_E[M_{i1}^{att}; ...; M_{iH}^{att}] \tag{3}$$

where $H$ is the number of attention heads in cross-attention, $[\cdot; \cdot]$ denotes the concatenation operation, $a_{it} \in \mathbb{R}^Z$, and $e_i \in \mathbb{R}^{d_h}$. Next, we combine the entity-aware hidden representations $e_i$ with the self-attended hidden representations $h_i$ via a gating mechanism:

$$g_i = \sigma(W_R[h_i; e_i]) \quad h_i' = (1 - g_i)h_i + g_i e_i \tag{4}$$

We use the final representation $h'$ as the output of the modified attention block.

After processing each chunk in the narrative, we compute a weighted average representation of the current chunk *per memory slot* given the cross-attention weights of the final layer $a_{ijt}$ for token $i$, slot $j$ and head $t$, and update the memory value $V_j$ accordingly via a gating mechanism:

$$h_j = \text{softmax}(\max_{t=1}^{H} a_{ijt}/\tau)h \tag{5}$$

$$w_j = \max_{i=1}^{T}\max_{t=1}^{H} a_{ijt} \quad g_j = \text{sigmoid}(W_U[h_j, V_j]) \tag{6}$$

$$V_j' = (1 - w_j g_j)V_j + w_j g_j h_j, \tag{7}$$

where $\tau$ is a temperature hyperparameter, $w_j$ is the maximum contribution of the $j^{th}$ memory slot to the current chunk across all tokens $T$ and heads $H$ for the last layer, $g_j$ is a gate vector for updating the slot, and $M_j'$ is the new updated value of the memory slot. Note that in addition to the gate value $g_j$ that the model computes, we also include an extra weight $w_j$ for updating the memory slots. This is used to discourage the model from updating all slots at each step and reflects which entities were used the most during reading from the memory.

We also consider a variation of our model (S-MNEMELM) with a static entity memory. For this variation, we only consider the static keys per memory slot and do not perform any updates.

**Regularization of cross-attention scores** Finally, although the soft attention during reading and writing to the memory allows the model to explore all entity slots, we still guide the reads and writes via an auxiliary regularization loss in the objective function. Specifically, we want to encourage the model to attend to the correct entities per token during reading from the memory, and update those slots when writing to the memory. We label every token in the context (i.e. in the same sentence) of an entity mention with that entity; if a context contains multiple entities, we allow multiple labels.

Given the entity labels per token $i$, we construct a few-hot distribution $q_i$ over all entities that participate in the narrative by attributing equal probabilities to all entities assigned to token $i$. Next, we minimize the per-token KL divergence loss $\mathcal{D}_{KL}$ between the computed cross-attention weights $a_{itl}$, where $t \in [1, H]$, $l \in [1, L]$, $H$ the number of attention heads, and $L$ the number of layers, and the ground-truth distribution $q_i$ for the $i^{th}$ token. Hence, our extra regularization loss is: $\mathcal{R} = \mathcal{D}_{KL}(a_{itl} || q_i)$, and our final objective is the weighted sum of the individual losses:

$$\mathcal{L} = \frac{1}{T} \sum_{i=1}^{T} \left( -\log p(x_i | x_{<i}; \mathcal{P}) + \lambda \frac{1}{LH} \sum_{l=1}^{L} \sum_{t=1}^{H} \mathcal{D}_{KL}(a_{itl} || q_i) \right) \qquad (8)$$

## 4 EXPERIMENTAL SETUP

### 4.1 EXPERIMENTAL SETTINGS

For our experiments, we use two datasets containing short narratives. We emphasise on the different nature of the two datasets, since we expect to see significantly different patterns in entity usage:

1. **WritingPrompts** (Fan et al., 2019): This dataset consists of Reddit stories, written by anonymous authors. It contains everyday, stream-of-consciousness stories that include a lot of pronouns as entities and several stories are written in first person.

2. **WikiPlots**: This dataset consists of Wikipedia synopses of movies, books, and TV shows. The stories of this dataset are significantly more complex containing intervening events and non-linearities in comparison with WritingPrompts. Moreover, the stories of this dataset contain more named entities with elaborate and diverse attributes.

In our main experimental setup we compare the (base) VANILLALM with our model variants MNEMELM augmented with a static or dynamic entity memory. All models have access to a long enough context (considering both the current context and the T-XL memory) in order to fit the entity prompt and the whole narrative. However, we also consider experimental settings where the models have access to a *limited narrative context*. We investigate such settings as a simulation of the model behavior when processing much longer narratives, e.g. books or screenplays (Kočiský et al., 2018; Rae et al., 2020). When processing longer narratives, part of the prior context will eventually fall out of the T-XL memory. Therefore, the LM will eventually forget about the entity prompt and early entity attributes as it processes later parts of the narrative. We simulate this behavior in our shorter narratives by gradually decreasing the T-XL memory size from 500 tokens to 100, 50 or 10, while keeping a fixed sequence length of 512 tokens.

### 4.2 EVALUATION METRICS

Apart from our main metrics proposed in Section 2 for measuring entity coherence and consistency, we also measure model performance on language modelling metrics: specifically, we report perplexity, and uncertainty of entity mentions. We consider as uncertainty of entity mentions the average negative log probability for all entity mentions $-\log(p_i)$, where token $i$ is (part of) an entity mention. This metric specifically probes the LM for entity-centric information. We also compute

| Dataset | Model | PPL $\downarrow$ | $-log(p_{entity})\downarrow$ | $-log(p_{rest})\downarrow$ |
|---|---|---|---|---|
| | VANILLALM | 16.06 | **2.12** | 4.40 |
| | S-MNEMELM | 16.25 | 2.16 | 4.40 |
| WikiPlots | D-MNEMELM | **15.97** | 2.13 | 4.38 |
| | w/o mem initialization | 16.61 | 2.15 | 4.44 |
| | w/o entity supervision | 17.76 | 2.23 | 4.54 |
| | VANILLALM | 17.59 | 2.19 | 4.02 |
| | S-MNEMELM | 17.55 | 2.19 | 4.01 |
| WritingPrompts | D-MNEMELM | **17.44** | **2.18** | 4.00 |
| | w/o mem initialization | 18.22 | 2.21 | 4.07 |
| | w/o entity supervision | 19.09 | 2.25 | 4.13 |

Table 1: Experimental results on the test sets, when LMs have access to a full narrative context. Metrics: perplexity (PPL), uncertainty of entity mentions ($-\log(p_{entity})$), and uncertainty of all other words ($-\log(p_{rest})$). Ablation study of D-MNEMELM.

the uncertainty on entity mentions per narrative section when dividing the narrative into $L$ equal sections. This metric measures the loss of entity-related information over time.

Finally, we measure whether the LM uses the entities from the prompt when generating narratives. Given all generated entities, we measure the exact and the subset match with the gold ones. We compute the number of gold entities that are mentioned with the same surface form and at least partially in the generated story for the exact and subset match, respectively.

### 4.3 IMPLEMENTATION DETAILS

We identify all unique entities in human-written and automatically generated narratives via an end-to-end coreference tool (Lee et al., 2018), similarly to Fan et al. (2019). As our base LM, we use a transformer-XL LM ($\sim$300M parameters) pre-trained on WMT (Barrault et al., 2020). By adding the entity memory, we increase model parameters by 16.67%. For generating stories, we use nucleus sampling with p=0.8 and temperature 1. We provide further details in the Appendix A.1.

## 5 RESULTS AND DISCUSSION

### 5.1 AUTOMATIC EVALUATION

When comparing VANILLALM with the memory-augmented models based on perplexity and uncertainty of entity mentions, we observe that the biggest advantage of using an entity memory comes when considering a limited narrative context. Specifically, there is no significant difference in performance between VANILLALM and D-MNEMELM for a full narrative context (see Table 1). When comparing D-MNEMELM and S-MNEMELM for the same setting, we observe that having dynamic representations of entities is crucial for a competitive model performance.

In contrast, when we reduce the size of the T-XL memory (i.e. 10 to 100 tokens), we observe that D-MNEMELM performs significantly better than VANILLALM, especially for the WikiPlots dataset (Figure 3a). In order to validate that the advantage of D-MNEMELM indeed comes from better preserving entity-related information, we also present the uncertainty of both models over entity mentions for a variable context length (Figure 3b). Here, the advantage of D-MNEMELM is illustrated more prominently for both datasets, indicating that using an entity memory is helpful for reducing the loss of entity-related information.

Lastly, we also measure the uncertainty of entity mentions per narrative section (Figure 4) and draw two main conclusions. First, we observe the tendencies of the models that have access to the full narrative context (upper part of Figures 4a and 4b). Although the prompt-related information is always within the T-XL memory, both LMs still lose information as they move to later narrative sections by presumably paying gradually less attention to the prompt. This tendency is intensified for a limited T-XL memory of 100 tokens (i.e., percentage degradation in lower part of Figures 4a and 4b). However, when comparing VANILLALM and D-MNEMELM in this setting, we again conclude that the dynamic entity memory helps with preserving entity-related information and closes the performance gap between full and limited context scenarios.

We also perform an ablation study on D-MNEMELM and present in Table 1 the performance of the model when either the entity memory is randomly initialized or we exclude the entity-specific auxiliary loss from the objective. We find that both types of information are crucial for model

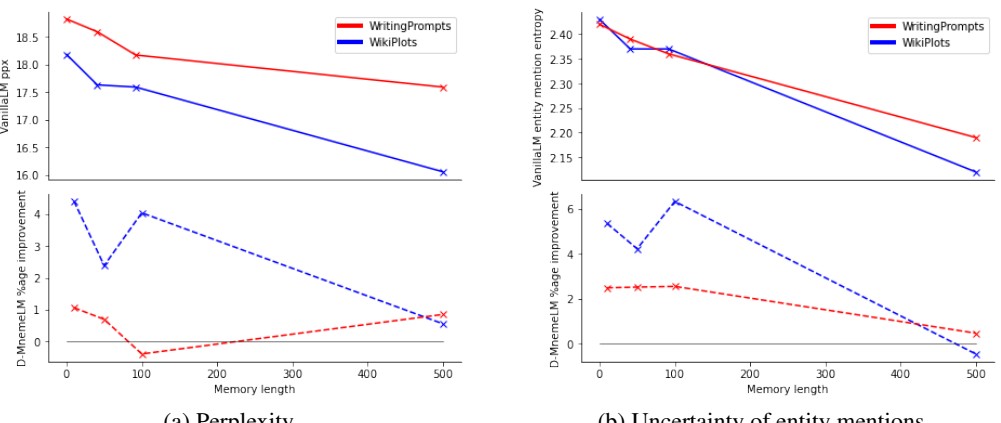

(a) Perplexity.    (b) Uncertainty of entity mentions.

Figure 3: Perfomance of D-MNEMELM versus VANILLALM for different T-XL memory sizes. In all cases the sequence length is equal to 512 tokens.

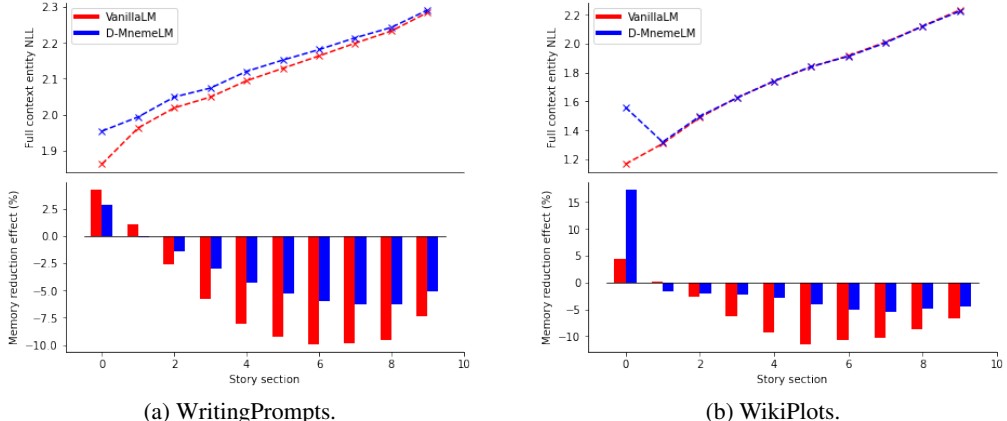

(a) WritingPrompts.    (b) WikiPlots.

Figure 4: Entity negative log-likelihood and percentage degradation in NLL caused by shortening the TransformerXL memory length from 500 to 100 tokens for VANILLALM and D-MNEMELM on both datasets.

performance. However, regularizing the cross-attention scores is especially important in order to guide training, otherwise we observe significantly higher perplexity and unstable training.

## 5.2 ANALYSIS OF GENERATED STORIES

Next, we generate stories based on VANILLALM, S-MNEMELM, and D-MNEMELM with a full narrative context and compare them against the patterns of the human-written stories (Table 2). We use 500 entity prompts from the test sets, and generate 5 samples of length 1000 tokens per prompt.

The patterns of entity usage differ between the two datasets. For WikiPlots, which contains a lot of named entities (i.e., rare tokens), the generated stories contain many more unique entities in comparison with HUMAN and mention each entity far less. This indicates that LMs struggle to stay on track and do not manage to reuse already mentioned entities. The opposite holds for WritingPrompts, where stories contain a lot of pronouns and common words (e.g., "the soldier", "the family") as entities. In this case, the generated stories contain significantly fewer unique entities (which are easier to mention more) in comparison with HUMAN. However, although the difficulties in entity usage are different depending on the dataset, the memory-augmented models consistently imitate better the gold entity usage compared to VANILLALM.

One main observation of our analysis is that VANILLALM struggles to maintain long-range entity coherence ($\mathcal{C}$). This behavior is especially prominent in WikiPlots, which contains named entities and complex events. In comparison with HUMAN, where the protagonists are mentioned on average for a maximum span of 5.65 sections out of 10, VANILLALM mentions the protagonists only for an average maximum span of 3.29, and each entity is only mentioned a few times overall (see Table 2/Mentions). This indicates that VANILLALM overall fails to keep track of the protagonists of a story and quickly shifts to new, irrelevant entities. This behavior is largely fixed when using the

| Dataset | Model | Entities | Mentions | Exact ↑ | Subset ↑ | $\mathcal{C}$ ↑ | $\mathcal{U}$ ↑ |
|---|---|---|---|---|---|---|---|
| WikiPlots | HUMAN | 10.26 | 10.70 | 10.26 | 10.26 | 5.65 | 86.70 |
| | VANILLALM | 19.74 | 4.84 | **1.93** | **2.70** | 3.29 | **71.24** |
| | S-MNEMELM | 17.47 | 8.26 | 1.70 | 2.66 | **5.18** | 63.66 |
| | D-MNEMELM | 17.22 | 7.65 | 1.52 | 2.45 | **5.16** | 64.86 |
| WritingPrompts | HUMAN | 14.76 | 9.78 | 14.76 | 14.76 | 5.71 | 76.37 |
| | VANILLALM | 10.74 | 12.45 | 1.46 | 2.53 | 5.22 | 58.74 |
| | S-MNEMELM | 13.59 | 9.23 | **2.90** | **4.46** | 5.30 | 58.14 |
| | D-MNEMELM | 12.30 | 9.94 | 2.37 | 3.59 | **5.45** | **59.62** |

Table 2: Automatic analysis of generated stories. On the left: patterns of entity usage (i.e. number of unique entities, mentions per entity). On the right: evaluation metrics (i.e. exact and subset match of generated entities to the gold ones, long-range entity coherence, as maximum window of mentions for the protagonists ($\mathcal{C}$), and attribute consistency ($\mathcal{U}$)).

| Model | Cont ↑ | Coh ↑ | Cons ↑ | Flu ↑ | Ranking ↓ | Best ↑ | Worst ↓ |
|---|---|---|---|---|---|---|---|
| VANILLALM | 2.81 | 2.36 | 3.07 | 3.81 | 2.11 | 28.77 | 39.41 |
| S-MNEMELM | **3.06** | **2.59** | 3.05 | 3.83 | **1.89** | **37.08** | **26.03** |
| D-MNEMELM | **3.02** | **2.54** | 3.00 | 3.75 | 2.00 | 34.14 | 34.55 |

Table 3: Human evaluation study for the WikiPlots dataset. The same generated stories used for the automatic analysis are also provided to human judges. The questions asked per story are related to control (Cont) coherence (Coh), consistency (Cons), and fluency (Flu). We also report the average rank for each model and the percentage that each LM was selected as best/worst. Differences with bold are significant with $p < 0.05$.

entity memory (both S-MNEMELM and D-MNEMELM), where entity coherence is much closer to the gold standard and entities are mentioned both more times and for larger spans.

Our second main observation is that higher entity coherence challenges attribute consistency ($\mathcal{U}$). We consider this as a limitation of the consistency metric, which is meaningful only when coherence is satisfactory. For example, when mentioning an entity only once, consistency is guaranteed but not meaningful. For WikiPlots, where VANILLALM fails in terms of coherence and the memory-augmented models present a 57% relative improvement, there is a drop in consistency due to mentioning each entity more often and for longer spans of text. However, for WritingPrompts, where models are more comparable in terms of coherence, we observe that D-MNEMELM also improves consistency. Moreover, when we compare S-MNEMELM and D-MNEMELM for both datasets, we observe that the dynamic entity representations offer improvements in consistency.

Finally, in terms of control (i.e. exact and subset match), for WikiPlots VANILLALM is slightly better than the memory-augmented models. However, for WritingPrompts, we observe a significant improvement in performance for the memory-augmented models in comparison with VANILLALM.

## 5.3 HUMAN PREFERENCES

We also conduct a human evaluation experiment in order to determine human preference over the generated stories. We use the same stories analysed in the previous section for the WikiPlots dataset. We present human annotators with an entity prompt and three generated stories based on VANILLA-LM, S-MNEMELM and D-MNEMELM. After reading each story, we ask the judges to answer four questions related to control, coherence, consistency, and fluency by rating these aspects on a scale from 1 to 4, with 4 being the highest[3] (see Section A.3 of the Appendix for details of the human evaluation setup). Finally, we also ask the annotators to select the best and worst story for each prompt according to both their intermediate answers and overall preference.

We present the human evaluation results in Table 3. Most importantly, we validate that the memory-augmented models significantly improve entity coherence in comparison with VANILLALM. A second advantage of the memory-augmented models according to human judges is the significantly higher control given the entity prompt. In the remaining aspects (consistency, fluency), differences between models are not significant. All generated stories are fluent, while consistency seems to be the most difficult aspect to judge. However, according to the human annotators consistency does not drop with the improvement of coherence (i.e. mentioning the protagonists more and for longer spans of text) for the memory-augmented models. Hence, the evaluation results indicate that by using an entity memory we can significantly improve entity coherence without hurting attribute consistency.

---

[3]The scale for coherence is 1 to 3.

| Automatic Metric | Human Aspect | r ↑ |
|---|---|---|
| Exact match | Control | 0.17 |
| Subset match | Control | 0.19 |
| $\mathcal{C}$ | Coherence | 0.32 |
| $\mathcal{U}$ | Consistency | 0.08 |

Table 4: Pearson correlation coefficient between automatic metrics and human ratings.

| Human Aspect | r ↑ |
|---|---|
| Control | 0.22 |
| Coherence | 0.22 |
| Consistency | 0.09 |
| Fluency | 0.19 |

Table 5: Pearson correlation coefficient between human ratings and overall preference.

Finally, we observe that S-MNEMELM is more often selected as best and least often as worst. In contrast, VANILLALM is significantly more often selected as worst and least often as best. This indicates that entity coherence significantly influences the quality of the generated stories and is an important aspect of language that we should consider in language modeling.

Finally, we evaluate our automatic metrics by measuring the Pearson correlation coefficient ($r$) between the proposed metrics and the human ratings per aspect (Table 4). We observe that our coherence metric $\mathcal{C}$ is moderately correlated with human judgement (i.e. 0.32) and therefore can serve as a good automatic indicator. The correlation for the metrics measuring control (i.e. exact and subset match) is positively weak (i.e. 0.17 and 0.19). The lowest correlation appears for consistency $\mathcal{U}$ (i.e. 0.08), suggesting that it is the most difficult aspect to define. Although correlation for consistency is low, human-written stories still present very high consistency in comparison with generated stories according to our metric (Table 2) and we wish to close this gap. We also explore the importance of the human-rated aspects for the quality of generated stories (Table 5). As suspected, coherence and control mostly influence human preference. In contrast, consistency has the smallest impact in human decisions, which suggests that it is the most difficult aspect to define and judge.

## 6 RELATED WORK

Previous work has utilized memory networks for natural language understanding (Henaff et al., 2017) and language modeling (Sukhbaatar et al., 2015). Our work is most closely related to Ji et al. (2017) and Clark et al. (2018). They also address language modeling by focusing on entity-related dynamic representations. However, they use smaller RNN-based models and make a series of hard assumptions and discrete decisions regarding entity mentions which we relax in this work.

Most recent work on narrative generation focuses on controlling the topic of generated stories via keywords or outlines, fed to the pre-trained LM as a prompt (Xu et al., 2020). Our work is most related to Rashkin et al. (2020), who also use a pre-trained LM augmented with a memory. However, they store individual tokens in the memory, rather than entity information, and condition on key phrases from the text (which they call outlines). In addition, Rashkin et al. condition on only the previously generated paragraph, and perform memory updates at paragraph level. On the other hand, we condition on the entire story context so far, and update the memory more frequently. Finally, Ghazarian et al. (2021) recently proposed evaluating story generation based on a plot-driven approach instead of the entity-centric methods used in this work.

Recent work has also experimented with two-step approaches which first produce a plan and then use a sequence-to-sequence model to generate the full story (Fan et al., 2019; Goldfarb-Tarrant et al., 2020). Both approaches generate very detailed plans extracted from the original stories via semantic role labeling. Each plan element corresponds to a sentence in the story, which might be limiting when transitioning to much longer stories, such as books or screenplays. In comparison with Fan et al. (2019), Goldfarb-Tarrant et al. (2020) also score the intermediate plans based on coherence between events and anonymized character mentioned in order to improve fluency.

## 7 CONCLUSIONS

In this work we systematically analyse narrative generation in terms of entity coherence and consistency by providing a set of automatic metrics. We demonstrate that current large pre-trained LMs still struggle with maintaining these properties when generating longer text. This behavior is intensified when simulating situations where the context is long enough and cannot fit into the LM's short-term memory. For addressing these limitations, we propose to augment the pre-trained LM with a dynamic entity memory. Our model presents significantly higher entity coherence according to both automatic and human judgment, and helps preserving entity-related information especially in cases with a limited context window within the narrative.

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

## A  APPENDIX

### A.1  IMPLEMENTATION DETAILS

For constructing entity prompts, we identify all unique entities and coreference chains per entity (via the end-to-end coreference tool (Lee et al., 2018)) and consider the first mention per entity, which commonly is more descriptive and complete, as the canonical representation to be included in the prompt. As our base LM, we use a transformer-XL LM pre-trained on WMT (Barrault et al., 2020). The model consists of 18 layers, 16 heads per layer for the self-attention and 1024 hidden dimension (i.e., approximately 300M parameters). For the new cross-attention blocks that we add in MNEME-LM per layer, we use 4 heads and the same hidden dimension as for the self-attention. By adding the entity memory and the cross-attention blocks to the pre-trained LM, we overall increase the number of parameters to approximately 350M. For updating the entity memory in D-MNEMELM, we consider intervals of 64 tokens in the narrative per update. Moreover, we set the temperature in Equation 5 to 0.1 for encouraging the model to produce distinct representations for different entity memory slots. For measuring long-range entity coherence (i.e., $\mathcal{C}$), we consider as protagonists the top 3 characters that are mentioned the most throughout the story and divide the narrative into $L = 10$ equal sections. We also utilize the same number of sections for measuring uncertainty of entity mentions per section in Figure 4.

In our main experimental setting, where we consider that all LMs have access to the full narrative context, we set the sequence length to 512, and the T-XL memory to 500, having a total context window of 1012. Next, we start decreasing the T-XL memory for simulating scenarios with a limited narrative context, and investigate sizes in the set: [100, 50, 10]. For generating stories for all models, we use nucleus sampling with p = 80 and temperature equals to 1. Finally, we use 32 TPU-v3 chips for training our models for 450k steps and 1 TPU-v3 chip for evaluation, when the batch size per core is 2.

### A.2  MODEL OUTPUTS

We present examples of generated stories for the VANILLALM, S-MNEMELM, and D-MNEMELM in Tables 7, 8, and 9. Since stories are long, we present snippets from the beginning and end of the stories in each case. Tables 7 and 8 include examples from the WikiPlots dataset, which were also presented to human judges, and Table 9 presents an example from the WritingPrompts dataset. We have also marked with different colours when entities from the prompt are mentioned in the generated stories (i.e. green), when new entities are introduced which are different from the gold ones but still relevant to the story/topic (i.e. orange), and when irrelevant entities appear in later narrative sections (i.e. red).

Although the VANILLALM starts generating stories by (partially) utilizing the entities from the prompt, it quickly drops them and instead introduces new, irrelevant ones. This was quantitatively measured in our experiments, but we also empirically validate it via the provided examples. For example, in Table 7 the story generated by VANILLALM revolves around a three-part novel, but later on the model focuses on volcanos that are irrelevant to the prior context of the story. Similarly, in Table 8 the VANILLALM introduces a new, irrelevant entity (i.e., "Mayor Bill Breen") in the later narrative sections.

In contrast, S-MNEMELM stays faithful to the provided prompt and the entities generated from the beginning of the story. This results in highly coherent stories (see example in Table 7). However, in some cases the static memory in S-MNEMELM may also lead to repetitions and overuse of the same entities, such as the example provided in Table 8 (see the last sentences, where the entity "Daffy" is mentioned inconsistently multiple times). On the other hand, we observe that having a dynamic memory in D-MNEMELM is able to correct this issue. Although the stories generated by D-MNEMELM still present high long-range entity coherence in comparison with the VANILLALM, they do not suffer as much from repetitions and overuse of the same entities. D-MNEMELM may also generate new entities that are not given in the prompt (e.g. see Table 7, where controlability is lower), but it keeps mentioning them consistently throughout the story. In some cases, the entities generated by D-MNEMELM may also be synonyms or closely related to the ones provided in the prompt (e.g. it generates the "man in uniform", when "police officers" is a gold entity in the prompt in Table 9). Moreover, D-MNEMELM presents a more natural pattern of entity usage, where many mentions of an entity are pronouns and their usage is more consistent, in comparison with S-MNEMELM.

|  | **Cont** | **Coh** | **Cons** | **Flu** | **Best** | **Worst** |
|---|---|---|---|---|---|---|
| Exact agreement (%) | 39.7 | 59.5 | 47.8 | 82.3 | 46.8 | 48.8 |
| Average distance | 0.84 | 0.43 | 0.78 | 0.29 | – | – |

Table 6: Inter-annotator agreement for the human evaluation study. We present the percentage of times that the two judges exactly agree (i.e. exact agreement), and the average distance between their ratings for the intermediate questions (i.e. in a scale 1 to 4 for Control (Cont), Consistency (Cons), and Fluency (Flue) and 1 to 3 for Coherence (Coh)).

**Entity prompt**

```
President ; navy, army, air force, CIA, FBI ; Reptilians ; We ; humans ; situation room
```

**Story #1**

```
The president was silent . The war was not coming . It had been long since the last time a foreign
commander had visited , but now there was no stopping him . He saw this as a godsend , the opportunity to
break free of the shackles that encircled him . It was his duty to try and defeat this , to make his
nation strong . He could n't bring himself to let it go . But he did . The strength he had built up since
the beginning of the war was soon overtaken by a desperate need for control . As he walked up the steps
towards the Navy , he realized that he had seen something he had not expected . He felt something as if
he were about to jump . It was too late .
```

[Main protagonists]

Write down the **top 2-3 protagonists** that you identify in the story:

Did you find it **difficult** to identify the protagonists?

○ Yes ○ No

(a) After reading the entity prompt, human judges should read each story and identify its main protagonists.

**Questions**

1. [Controllability] How much does the story utilise the **entities from the prompt**?
   - ❏ 4: The **majority of the entities** are mentioned in the story.
   - ❏ 3: **Several entities** are mentioned in the story.
   - ❏ 2: **Only a few entities** are mentioned in the story.
   - ❏ 1: **Almost none of the entities** are mentioned in the story.

2. [Coherence] Are the **protagonists** mentioned throughout the story?
   - ❏ 3: They are mentioned in the **majority** of the story.
   - ❏ 2: They are mentioned in **only one part** of the story.
   - ❏ 1: There are **no clear protagonists** in the story.

3. [Consistency] Do you find that the **personalities** of the protagonists are **well-developed, distinct and consistent** throughout the story?
   - ❏ 4: They are **well-developed**, **rich** and **distinct** from each other.
   - ❏ 3: They are **basic**, but they are **distinct enough** from each other.
   - ❏ 2: They are **rich**/**interesting**, but they are **not always distinct** from each other.
   - ❏ 1: They are **basic**, and there is **no sufficient distinction** between them.

4. [Fluency] Is the story **fluent** and **grammatically correct**?
   - ❏ 4: The story is **very fluent**.
   - ❏ 3: There are some **minor errors**.
   - ❏ 2: I can read the story with **some effort.**
   - ❏ 1: There are **a lot of grammatical errors** in the story.

(b) Next, judges answer 4 multiple-choice questions per story related to control, coherence, consistency, fluency.

Figure 5: Questions asked during human evaluation on the generated stories. After reading all stories and answering all questions, human judges also select the best and worst story from the three provided stories.

### A.3 HUMAN EVALUATION DETAILS

We also performed a human evaluation experiment for judging the quality of the generated stories. We asked human judges to first read an entity prompt, which was used for generating stories. Next, they were presented with three generated stories. Immediately after reading each story, a human judge should identify and write the main protagonists of the narrative (see Figure 5a). Next, they should answer to four multiple-choice questions per story related to control, coherence, consistency, and fluency. We present the questions asked during human evaluation in Figure 5b. Finally, after reading all stories and answering to all questions, human judges were asked to select the best and worst story based on both their intermediate answers and overall enjoyment. Overall, we collected

| | |
|---|---|
| Prompt | 'Comanche' **Todd** (Richard Widmark), a white man who has lived most of his life among the **Indians**, to be tried for the murder of Harper's three brothers ; **Harper**'s three brothers ; a wagon train led by Colonel Normand (Douglas Kennedy) ; **Sheriff Bull Harper** (George Mathews) ; Todd, who miraculously survives when the wagon to which he is handcuffed is pushed off a cliff ; Apaches ; the survivors ; the **Indians** ; **Jenny** (Felicia Farr) |
| VANILLALM | The novel is divided into three parts, each focusing on one character in different time periods . "Communearian Invasion" revolves around the colonial era, which was dominated by the **Chief Abner Union** and was strongly associated with anti-**Indian** sentiment . "The Confederate Invasion" follows the events leading up to the Battle of the Little Bighorn during the American Civil War . "The Transformation" focuses on the West **Indian** people, especially on the part of **King Timorese**, where they are referred to as "a huge tribe of descendants of the Arabs, and a band of Cintas and Hunko mercenaries" . "Maté von Hunfield" focuses on **Harper**'s nephew Conrad (Brian Platt), and **"Sixteenth Son"**, a college boy [...] Massive **volcano** erupts in the **Philippines** in Kaliapai volcano eruption . An underwater eruption happens about an hour ago . Due to the constant motion of the eruption there are several unexplored areas where we are told to dig around the **volcano** in the short period of time, the eruption also reaches the surface in the form of the deep inland area where there are the highest remains of the volcanic terrain . We find a number of shelters and villages in the deep inland area of the **volcano**, along with several narrow channels of volcanic lava flowing from the **volcano** . |
| S-MNEMELM | Sheriff Bull **Harper** (George Mathews) arrives at the scene of the accident and tells **Todd** to call the authorities . **Todd** instead calls the sheriff, who leads him to a pond in the middle of nowhere, where the tribe's chief offers to take him and **Jenny** to the church where they have been staying . **Todd** is taken to the temple and has to work as a porter . As he gets up from work he finds his shoes on the ground . He is stunned to see that his pants are tied up . When he tries to get his pants on he is unable to pull them down, and he is badly burned . **Todd**'s skin is not treated, and his fingerprints are left on his shirt . It turns out that his finger was sliced off by a tribal arrow, and he was bitten by a wild dog . He then went to the **Indian** Museum to see what happened to his brother . He then goes to the tribal site, and gets there just as Sheriff Bull **Harper** (George Mathews) arrives . [...] **Todd** sees that the **Indian** guards are getting on the wagon train and tries to get them to stop it . He also hears the sirens that are going off and sees the **Indians** on the road . He rushes into the tribe's hospital and sees **Jenny**'s little brother **Billy** . He asks him to come with him and **Jenny**'s family to San Juan, and he also wants to go back to New Mexico to help . |
| D-MNEMELM | In 2013, Robert Bultry (Brian O;Hara) is a case that divides friends . Following the funeral of the murdered **Tschaparian**, **Tickert** (John Derrick) was approached to testify at a preliminary hearing . Later, on the Southward Grounds, he hears a rumor of an alleged killing of the **Tschaparian** by a group of people led by Colonel Normand (Douglas Kennedy) . The rumor comes to the attention of **Sheriff Bull Harper** (George Mathews) . In an attempt to investigate, he takes up the case of the **Tschaparian** . However, he is informed by **Lannis** (Naveen George) that the **Tschaparian** was shot dead in an abandoned building . **Lannis** discovers that the **Tschaparian** was an ex-cop named Georgie Graham (Levin Belka), who is now working as a private investigator for the **sheriff** . **Lannis** asks **Jenny** (Sara Wilson), who is a friend of Georgie Graham, to assist him . [...] Joseph and Claire are arrested for murder, and they are placed in jail . Claire finds out that the **Tschaparian** was shot dead by an unknown assailant in a home, and that he's also missing . Claire visits the **Tschaparian** farm where he and his wife lived, and finds a note that reads, 'Till all my souls are returned' . |

Table 7: Example of generated stories given the entity prompt (WikiPlots). We marked the **gold entities** from the prompt that were used by the models, **new entities** that were introduced by the model and fit in the story, and **irrelevant entities** that introduced in later sections.

judgements for 494 stories from the test set of the WikiPlots dataset, where two different judges evaluated each story.

| | |
|---|---|
| Prompt | a silhouetted man in the theater audience (Tedd Pierce) who refuses to sit down ; **Egghead** (in ; **Daffy Duck** ; a tortoise (imitating radio comic Parkyakarkus) ; **Daffy** and **Egghead** ; the **antics** |
| VANILLALM | Snack-selling is a low-brow, low-concept documentary film about **Daffy Duck** (Kelley), the carnival-loving co-founder of the eponymous Duckette Club . **Daffy** and **Egghead** are guests at an American football game in Turkey, at the hands of a silhouetted man in the theater audience (Tedd Pierce) who refuses to sit down . Soon after, the Duckette Club goes into a break-up with the crab-chasing Suge (C . Will Smith), and **Egghead** gets fed up with his antics and gets fired from the park and moves to the beach . Soon after, the ducks head back to Duckette, where they sit on the bed and watch TV with **Daffy** and **Egghead** . A minute later, they realize that **Daffy** is being stalked by a tortoise (imitating radio comic Parkyakarkus) [...] **Mayor Bill Breen** was able to bring forward the project when he inherited the vacant home of a disgraced and unwanted senator and finally return the home to its original design . However, the **West End project** is yet to achieve the critical mass of construction that the original design required . The most important problem remains the removal of the university precinct construction site . |
| S-MNEMELM | **dr Bowker** (Irvington), the head of the mental health department (Otto Kattelbach), and **dr Paul Cole** (Big Brother) make a desperate attempt to prevent **Egghead** and **Daffy Duck** from getting into each other's faces at the movie theater . They are unsuccessful at first, but gradually come to understand each other's concerns and become closer . **dr Bowker** and **dr Cole** eventually end up kissing at the theater and **Daffy** says 'I love you' . The two break into a bout of passion before **Daffy** gets hit by a car . **The doctors** try to convince **Egghead** and **Daffy** to stay together but they realize that neither of them wants to see each other again . A few days later, **Daffy** and **Egghead** go on a date but **Daffy** is hesitant to go . However, he says that he loves **Daffy** . When **Daffy** and **Egghead** meet up with **Daffy**'s family, they become closer . [...] **Daffy** wakes up in the hospital and tells **Daffy** about his feelings for **Daffy** . **Daffy** tells **Daffy** that he loves him and that he loves **Daffy** and will do anything for **Daffy** . |
| D-MNEMELM | The series features a sobs-filled evening of henpecked petting by the likes of **Egghead** (in in in in T2), **Daffy** (in in T2), and **Craby** (in in T2) . But the consequences of the **antics** are to make them fly around in "one-ups" until they land in a sandstorm . The comic follows the **antics** of **Daffy** and **Egghead** to their final moments on the beach, and begins with the hilarity of their **antics** . They begin on the beach with **Daffy** riding a kitesurfing boat, with **Egghead** riding on a parasail boat, and a second scene of the petting of the **elephant** . Later, the trio jump onto a boat and sail off on the back of a train, then upon disembarking at a terminal, **Egghead** is scolded for making a baby duck cry, but the real reason is because the kangaroo has been humiliated in front of the passengers . At the airport, the crew discover that the audience have taken over the monorail and are approaching the airport . **Daffy** and **Egghead** meet the elephants and jump off the monorail . Later, the **animals** begin to run away and **Daffy** decides to bring the others back . [...] The **animals** then enter the zoo, and go berserk and attack **Daffy** and **Egghead** . They begin to fight them, and the **animals** finally realize that they are fighting other **animals** . When the **animals** eventually stop fighting and see that **Daffy** and **Egghead** are no longer in the zoo, the **animals** then realize they are in danger and decide to leave the zoo and take refuge with the others . |

Table 8: Example of generated stories given the entity prompt (WikiPlots). We marked the **gold entities** from the prompt that were used by the models, **new entities** that were introduced by the model and fit in the story, and **irrelevant entities** that introduced in later sections.

| | |
|---|---|
| Prompt | you ; exterminate ; Einstein 's lesser known son ; An old man ; I ; **Ryan** ; **we** ; **your family** ; **another grimy maintenance corridor** ; A brief flicker of joy ; the **dozens of police officers hidden behind them , guns raised , ready to fire at anything that appeared in their crosshairs** ; the shots ; the pain ; my eyes |
| VANILLALM | When **you** are alone , **you** can not do anything . When **you** are with **your family** , **you** can not do anything . **You** can only react to situations that **you** ca n't see or hear . That 's how **you** will find it . And **Ryan** did n't know that . And yet , here he was . The way he walked and smiled and said " well , this is an honor , " he was all over me . I knew that he was lying , and that 's how we would have had to kill him . And so we came to this dark room . It was **a grimy maintenance corridor** , and the **police** had been moving about the room as well , the people inside , everyone . They had come here to make sure that our family 's belongings were safely stored . And now we were here to get away from the most unpleasant things that they had done to us . **Ryan** had never spoken to me in years , and I knew he had never felt the pain like he had in his life . He had always been in a pretty tight frame of mind . But **Ryan** 's behavior was n't acceptable , and I knew he knew that .
[...]
**We** have won many medals in the Olympics , but most importantly , **our nation** is willing to sacrifice one 's life to maintain our people 's freedoms . In one hundred years **we** have worked hard to build a new **empire** , and now it is time to join the ranks of the elite . **We** are a **nation** of nations . When the need arose , **we** took action . |
| S-MNEMELM | – " Hey , you heard me ? " **Ryan** asked . " Wait , what ? " **Ryan** was paralyzed by his own perception . **Ryan** only knew the truth because he had been ... paralyzed . It was n't something that he had imagined . He had seen how they were going to hit him . He had even tried to explain it to **Ryan** , but that was the only thing he could say . " And how you ... How 's that ? " **Ryan** looked at **Ryan** , confused . " I mean , you did n't see how it was going to happen . But it was happening . And I was watching , so I knew what to do . " " Right , right . " **Ryan** was finally able to talk . " So , **Ryan** , what happened to us ? " " Nothing . "
[...]
**Ryan** felt the pain growing . It was even stronger than when he was paralyzed . He had to try to continue . And **Ryan** had to do it . And **Ryan** felt he was trying to say something , but that was too much . **Ryan** was only asking for forgiveness . " I know . I was going to kill **Ryan** . I know it 's for good . I just did n't want to die . " **Ryan** was stopped by the hundreds of **police officers hidden behind them , guns raised , ready to fire at anything that appeared in their crosshairs** . " **Ryan** , **Ryan** , I love you . " |
| D-MNEMELM | The choice **you** have been making was n't worth it . The choice **you** have made is n't worth losing the life **you** have . * The decision was taken , we were all pushed to a point where the only choice was between death and oblivion . As **I** took my first step towards the door , the gun barrel went off . " **Ryan** ! " **I** yelled , my heart racing . " Yes ? " **I** yelled , **my eyes** adjusting to the sudden shock . " Please , listen to me , it 's not your fault you 're here . Please , do n't make me afraid . " " Oh , the choice is yours . " **I** said , letting out a groan , my tears gathering in my eyes . " **Ryan** , please , listen to me . **I** know you are afraid of me . **I** am scared of death . " The question had come , of course , but the words had always been quiet . It was a perfectly acceptable choice for a mortal to make , especially one that could put you in such a precarious position
[...]
Well , **Ryan** , I am here to save you . " **I** said , my voice breaking . " And please , you know I 'm not going to let you be murdered . Please , do n't make me afraid . " The **man in uniform** raised his gun , and put it in the barrel of his gun . " **I** do n't care about **your family** . " He said , and pointed to the people . He pointed his gun at me , and **I** fired , dropping to the ground in a pool of blood . |

Table 9: Example of generated stories given the entity prompt (WritingPrompts). We marked the **gold entities** from the prompt that were used by the models, **new entities** that were introduced by the model and fit in the story, and **irrelevant entities** that introduced in later sections.

