# OpenReview forum: "Towards Coherent and Consistent Use of Entities in Narrative Generation"
_ICLR.cc/2022/Conference — ICLR 2022 Submitted_

### Official Review · Reviewer_Q7Fu · 2021-11-03

**Correctness:** 3
**Technical Novelty And Significance:** 3
**Empirical Novelty And Significance:** Not applicable
**Recommendation:** 6
**Confidence:** 4

**Main Review:**

Strengths and Weakness:
1. The proposed approach for tackling entity consistency and coherency through the use of memory structures that augment language modeling is novel and interesting. Further, the dynamic nature of the proposed entity memory network allows multiple attributes to be associated with an entity.
2. One of the contributions of this work is the proposed evaluation metric for entity coherence and consistency. The metric for entity coherence based on calculating C_i as the difference in the indices of first and last mentions does not truly capture coherence. Further, the example provided in the paper Fig 1b seems to be an issue with local coherence instead of long-range entity coherence.
3. With regards to entity consistency, the authors propose quantifying it through the use of attributes associated with each entity is interesting. However, this approach does not take into consideration multiple entities in a sentence and attribution association in those cases.
4. Evaluation is done on the standard datasets of writing prompts and wikiplots. However, it would have been better if the authors had used more baselines to compare their proposed approaches instead of a simple LM for both datasets.  From the automated metric perspective, the proposed addition of entity network seems to be performing worse than a vanilla LM on the wikiplots datasets on the metrics as shown in Table 1. This trend of vanilla LM outperforming the proposed approach is evident in the proposed automated metrics in Table 2.
5. The proposed automated metrics when compared to the human evaluation show a low to moderate correlation to human judgments. The correlation for consistency is pretty low which raises questions about the proposed metric and its effectiveness to capture entity consistency.

Questions:
1. With regards to entity consistency, how were the attributes assigned if there are multiple entities mentioned in the sentence with attributes?
2.  Why does the attribute consistency on the wikiplots dataset using a simple vanilla LM outperform the proposed entity memory network augmented LM?
3. why is the scale for coherence different from the other metrics?
4. How many human annotators evaluated the generated outputs? What is the IAA?

**Summary Of The Paper:**

The authors of this paper propose to tackle the problem of coherency and the consistent use of entities for the task of narrative generation. To achieve this, the authors propose an approach of augmenting pretrained language models with entity memory and cross attention blocks that places an emphasis on the entities in the narrative. The proposed approach is evaluated on the WritingPrompts and the WikiPlots datasets. From an evaluation perspective, the authors propose two new evaluation metrics for measuring entity coherence and consistency. To measure entity coherence, the authors propose dividing the narrative into L equal sections and computing the maximum interval span of each entity as the difference between first and last mention respectively. Similarly for entity consistency, the authors propose measuring entity consistency through the concept of measuring attribute consistency within the narrative.

Contributions:
1. A new approach of augmenting language models with entity memory that helps in entity consistency and coherence across narrative generation tasks
2. Two new metrics to evaluate consistency and coherency in the generated outputs.


**Summary Of The Review:**

This paper tackles the problem of entity coherence and consistency in narrative generation and tackles the problem through the use of an entity memory network that augments the pretrained language model. This proposed approach is interesting and demonstrates the effectiveness of the proposed approach on two datasets. The clear gains of the proposed approach can only be seen on one of the datasets compared to a simple baseline. Further, the authors also propose using two new metrics for automated evaluation. These metrics show a low to moderate correlation to human judgments.

---

> ### Author Response · Authors · 2021-11-16
> **Response to Reviewer Q7Fu**
>
> Thank you very much for your detailed review and constructive feedback. We hope we can address your concerns:
>
> **Review point 2**: Please see general response 1a.
>
> **Review points 3 and 5, Questions 1 and 2**: Please see general response 1b.
>
> **Review point 4**: Please see general response 2. Moreover, regarding more baselines, please see response 2 to Reviewer 2yed.
>
> **Questions 3 and 4**: Please see general response 3.

---

> ### Author Response · Authors · 2021-11-30
> **Follow-up**
>
> Dear reviewer, we hope you've had the chance to take a look at our response and paper revision. Please let us know if the response addressed your concerns or there is anything else that is still unclear. We are happy to provide further clarification.
>
> Thank you for your time!

---

### Official Review · Reviewer_XC2i · 2021-11-03

**Correctness:** 4
**Technical Novelty And Significance:** 4
**Empirical Novelty And Significance:** 3
**Recommendation:** 5
**Confidence:** 5

**Main Review:**

Pros:
The motivation of the work which is relevant to a well-known issue in existing long text generative models has been written clearly and figures are very helpful in this regard. Author try to contribute in two main problems first proposing automatic metrics to measure and analyze the coherence of generated texts by existing LMs and second proposing a model to improve the coherence of generated texts by augmenting an entity memory. Different experiments and ablation studies show the effects of each component in the model from different perspectives.

Cons:
There are some not clear sections that need to be explained more clearly. Here are a set of questions and comments that addressing them can be beneficial in better understanding the paper.

Main question is regarding different forms of the same entity and how they are handled in the model. Assume we have one entity but in two different forms such as mom and mother throughout the plots, according to what has been explained in the paper model considers them as two separate entities. If this is true, what is the impact of language different forms of words on the model's performance?

For automatic evaluation of coherence of texts the C metric only considers the first and last happening location of the entities in the text, while this is very abstract level and can not be a good and fair representative of the coherence in the text level. There are some recent works that try to assess the coherence of the generated narratives by using the plots and manipulating them (Ghazarian et al. 2021). In order to show the positive impact of proposed evaluation metric it is recommended to compare with more recent works.


In the method section, since the main idea of the work is about entity memory, presenting some instances for plots/entities/attributes  can help author to better follow the steps and the proposed model.

According to table 1, if the model can have access to the whole context then there is not that much difference between the proposed model and the vanilla LMs? Therefore if we use LMs that can process long texts such as longformer then we won't have benefits coming from the entity coherence proposed by this model?

For human evaluation, how do you ensure the quality of the annotations are good? what are the agreements between human judgments? how many annotations have been collected for each narrative?






**Summary Of The Paper:**

In this paper, the main contributions are 1)analyze the entity coherence of  pretrained LMs by leveraging some existing metrics and also four newly proposed metrics 2) propose a new generative model to have a higher quality generated narratives from entity coherence and consistency perspectives. Authors develop a model by adding a dynamic entity memory to the existing pretrained LMs. The cross attention between input text and entity memory alongside the self-attention coming from pretrained-LM helps to not forget the previous entities and their attributes during generating narratives.

**Summary Of The Review:**

In summary the idea of using entity memory and computing cross attention between input and this memory to incorporate information from entities and their attributes looks promising. However, the metrics showing the coherence of the text is only limited to the location of the entities while for the text the overall coherence of the text should also be considered. Some sections are opaque but I think if authors can address those the paper can be effective for the community.

---

> ### Author Response · Authors · 2021-11-16
> **Response to Reviewer XC2i**
>
> Thank you very much for your detailed review and constructive feedback. We hope we can address your concerns:
>
> **Regarding different forms of the same entity**: We use an end-to-end coreference resolution tool (Lee et al., 2018) as part of the preprocessing of the stories. Based on this tool, we identify all unique entities in the story and all mentions per entity. Hence, given your example (mother vs. mom), the tool should categorize these two forms as mentions of the same entity. If so, our model then learns to treat these mentions as the same entity. However, since the identification of entities and mentions per entity in this work is automatic, there might be mistakes.
>
> **Regarding entity coherence**: Please see general response 1a. Moreover, thank you for the suggestions and the reference provided. In this work, we specifically focus on entity-related coherence, analyse stories centered around entities and investigate the patterns of entity usage in human-written versus automatically generated narratives. However, it is true that there are two different directions/theories regarding story generation and analysis: entity-based versus plot-based. Although we focus on the former, we will include the given reference as an example of the later direction.
>
> **Regarding the models that have access to full context**: When we consider the full narrative context, we indeed do not observe any difference in perplexity between the two models (Table 1). However, perplexity is not enough for judging the quality of generated text. When analysing the models by generating stories and computing a series of metrics, we observe that our proposed model presents significantly higher entity coherence. Note that all results presented in Tables 2 and 3 including human evaluation are based on the model variants that have access to the full narrative context. However, in any case it would be interesting to test the limits of models such as the Longformer, since recent works demonstrate that even these models cannot effectively process really long context (Sun et al., 2021).
>
> **Regarding human evaluation**: See general response 2
>
> (Sun et al., 2021) Sun, Simeng, et al. "Do Long-Range Language Models Actually Use Long-Range Context?." Proceedings of the 2021 Conference on Empirical Methods in Natural Language Processing. 2021.

---

> ### Author Response · Authors · 2021-11-30
> **Follow-up**
>
> Dear reviewer, we hope you've had the chance to take a look at our response and paper revision. Please let us know if the response addressed your concerns or there is anything else that is still unclear. We are happy to provide further clarification.
>
> Thank you for your time!

---

### Official Review · Reviewer_2yed · 2021-11-03

**Correctness:** 3
**Technical Novelty And Significance:** 2
**Empirical Novelty And Significance:** 2
**Recommendation:** 5
**Confidence:** 3

**Main Review:**

Strengths:
1. The authors address important problems of LMs on entity coherence and consistency when generating long stories.

2. The work proposes new automatic metrics for analyzing narrative generation where some show high correlation with human evaluation.

3. The work proposes an improved pre-trained LMs for narrative generation by adding an entity memory which shows competitive results compared with the Vanilla LM.

4. Extensive results discussion, evaluation, and ablation analysis are presented to clearly show the performance of some of the proposed techniques.

Weaknesses:
1. The codes are not made publicly available for reproducibility.

2. The proposed entity memory-based pre-trained LM is only compared with Vanilla LM. It would be more interesting to show how the proposed model performs when compared with other state-of-the-art LMs such as BERT and the main prior related works of Clark et al. (2018) and Ji et al. (2017) that have been discussed in the introduction section.

3. There is a lack of experimental support of why the proposed entity consistency metric is  needed since it gives lower correlation with human ratings in Table 4. It would be better to add more discussion on why it can still be considered to be adapted.


4. Minor weakness:
Figure 1(a) is not easy to understand. The <sep> tokens used do not clearly show where the entity starts and ends. It is suggested that you prefer using <sep> and </sep> pair, where the former shows the beginning of the entity and the latter shows the end.

**Summary Of The Paper:**

Considering little analysis that has been done on analyzing the ability of pre-trained language models (LMs) to maintain entity coherence and consistency, the authors take narrative generation as a case study and analyse the long-range entity coherence and consistency in generated stories. They firstly quantify the limitations of current LMs by proposing a set of automatic metrics for measuring model performance in terms of entity usage, and propose an augmented pre-trained LM with a dynamic entity memory in an end-to-end manner by using an auxiliary entity-related loss for guiding the reads and writes to the memory, which increases both entity  coherence and consistency in the generated stories.

**Summary Of The Review:**

Overall, the paper is well written and organized. The studied problem is very important which makes the paper to have many audiences at ICLR and the proposed techniques can benefit the field however after performing few more experiments to support them (see Weaknesses section Point 2 and 3).

---

> ### Author Response · Authors · 2021-11-16
> **Response to Reviewer 2yed**
>
> Thank you very much for your detailed review and constructive feedback. We hope we can address your concerns:
>
> 1. Our code heavily relies on internal (proprietary) infrastructure, and this is why it is challenging to directly open source it. However, we tried to include as many details of our implementation as possible in order to guarantee reproducibility. Moreover, we will try to release our code upon acceptance for other researchers to use.
>
> 2. We do not compare against prior work (Clark et al., 2018; Ji et al., 2017) for two reasons. First, these works are based on small RNN-based models that perform far worse than current large pre-trained LMs. Therefore, the comparison would not have been meaningful or fair. Second, it was impossible for us to re-implement these works with the state-of-the-art LMs, since there are incompatibilities between the assumptions made in these works and the current assumptions for the transformer-based models. For example, Clark et al., 2018 and Ji et al., 2017 predict whether to mention an entity, which entity to mention, and whether to update the dynamic representation of an entity at each time step, which is against the parallel architecture of the transformer (i.e., it would prohibitively slow down training). These incompatibilities are part of our motivation for proposing our approach. Moreover, BERT is a language representation learning model, and is not designed for language generation; however, parameter-wise, our VanillaLM has 300M parameters, which is comparable to BERT and other models from the literature.
>
> 3. See general response 1b.
>
> 4. Thank you for the suggestion in making our example more clear, we will update the paper accordingly.

---

### Official Review · Reviewer_sZJR · 2021-11-04

**Correctness:** 3
**Technical Novelty And Significance:** 2
**Empirical Novelty And Significance:** 2
**Recommendation:** 5
**Confidence:** 4

**Main Review:**

Strengths:
(1) The work is to address a challenging and practical problem in generating long narratives.
(2) The authors proposed two intuitive metrics to measure the entity coherence and consistency based on their mentions and context, which are beneficial to the following research.
(3) The proposed entity memory augmented model is reasonable and improved entity coherence/consistency in some of the experiments. The authors also provided a detailed analysis of the models in various settings.

Weakness:
(1) The two proposed metrics about entity coherence and consistency are intuitive but not rigorous. The entity coherence is based on the occurrence of the entity mentions across different sections. However, the coreference is accurate especially when the context is wide, and even an entity is mentioned across a wide context, there are a lot of other factors to consider to quantify the coherence, e.g., the intermediate mentions used in-between the first and the last mention. Similarly, for consistency, the authors considered all verbs and adjectives as the attributes of the entity. What's the reason? and how can it measure the inconsistency in the example of Figure1 (b)?
(2) The model design is reasonable, but it depends on several key hyperparameters, e.g., the size of the entity memory or the context memory, the length of each narrative section. They may have much impact on the final performance and on different datasets, they should have different choices. Also, it seems the entity set is given to initialize the memory, then what if the model generates new entities?
(3) The results didn't consistently demonstrate the improvement of the proposed approach, e.g., in Table 2, on the WikiPlots dataset, the VanillaLM is better than other models for most of the metrics; and on the WritingPrompts dataset, the VanillaLM seems to generate more mentions in average.
(4) The analysis section is very difficult to follow. Some analysis is just based on wordy descriptions without details or examples. For example, in the automatic evaluation section, it's not clear how the uncertainty is computed, how to interpret the Figures, especially Figure 4, what does the control in section 5.2 mean? and what are the detailed criteria for human rating (1-4)?

**Summary Of The Paper:**

This work focuses on improving the long-range entity coherence and consistency when generating long narrative stories. The authors proposed two metrics to measure the entity coherence and consistency in terms of entity usage in generated narratives. They also propose to augment a pre-trained LM with a dynamic entity memory and cross-attention to improve the entity usage.

**Summary Of The Review:**

Overall, this work proposed some interesting ideas to analyze and improve the entity usage in long narrative generation task. However, the metrics need to be carefully designed, and the analysis needs to be improved.

---

> ### Author Response · Authors · 2021-11-16
> **Response to Reviewer sZJR**
>
> Thank you very much for your detailed review and constructive feedback. We hope we can address your concerns:
>
> 1. Please see general response 1
>
> 2. In this work, we mostly focus on how we can successfully add entity-related information to a pre-trained LM with a limited size entity memory. In order to examine this question, we make the assumption that all entities are given.  During inference, the model might generate new entities that are not in the entity memory. Although we have an extra slot in the entity memory for accounting for information that does not correspond to any of the given entities, the model should ideally be able to add a new slot in this case. This is a limitation of our work, which we leave for future research (i.e., converting this model to an unconditional one).  We also experimented with different hyperparameters for the model per dataset. We find that having a maximum size of 20 entities in the memory is sufficient for both datasets, since they rarely contain more entities in the stories. Furthermore, regarding the different sizes of context windows, we conducted an extensive ablation study of how performance is affected when we consider a prior context of different size (see Section 5.1 and Figures 3,4). The main conclusion of this study is that adding the entity memory makes the model more resilient in entity-related information loss when we consider prior context of variable length.
>
> 3. Please see general response 2
>
> 4. We compute uncertainty as the negative log likelihood of all entity mentions. Higher negative log likelihood for tokens that are part of an entity mention signals that the model has a high uncertainty about the entity that it should mention next. Figure 4 shows this uncertainty per narrative section. In general, both models start from a very low uncertainty in the first section, since they are given the entity prompt. However, as the story evolves and more prior context is added to the LM’s memory, the uncertainty for the entity mentions becomes higher. For the models with access to a limited prior context, this behavior is intensified further (lower part of Figure 4 shows the percentage degradation) and we can see that they lose entity-related information as they move to later sections. However, when comparing the VanillaLM and our model, we observe that the entity memory reduces the information loss, especially for the later narrative sections. Regarding human evaluation, please also see general response 3.

---

> ### Author Response · Authors · 2021-11-30
> **Follow-up**
>
> Dear reviewer, we hope you've had the chance to take a look at our response and paper revision. Please let us know if the response addressed your concerns or there is anything else that is still unclear. We are happy to provide further clarification.
>
> Thank you for your time!

---

### Author Response · Authors · 2021-11-16
**General Response**

We would like to thank all reviewers for their valuable feedback. We try to address all their concerns and will update our paper accordingly. We note that many of our reviewers had similar concerns, and we present a general reply here:

1. **Automatic Evaluation Metrics**

    a. **Entity coherence**: During preliminary experiments, we considered several different metrics for quantifying entity coherence. We started our analysis from local coherence metrics that were previously suggested in literature, such as distance-based clustering coefficient (DCC; Lioma et al., 2016), local entity coherence (Barzilay et al., 2008), and distance-based linkage coherence (DLC)(Mesgar et al., 2014; Guinaudeau et al., 2013). These metrics measure the patterns of entity usage in documents divided into sentences based on grids or bipartite graphs. However, our analysis demonstrated that large LMs perform on par or better compared with humans based on these metrics. On the other hand, when we empirically investigate the generated stories, we discover that LMs actually still suffer from low entity coherence. One major problem when generating long enough stories (e.g., 1000 tokens) is that the LM may use the same entity more than once locally, but it tends to forget about the initial entities and instead introduce new irrelevant ones in the long run. We present examples of this phenomenon in the Appendix (Tables 7, 8, and 9). Our proposed metric therefore aims to capture this pattern of entity usage. We experimented with variants of the metric, where instead of considering only the first and last mentions we also considered all intermediate mentions as well. Specifically, we found that to capture the main problem of modern LMs, it is important to divide the story into broader sections and investigate the entity usage per section instead of focusing on sentence-level analysis as in previous approaches. Then, we measured the maximum window of mentions for an entity as the number of sections (which is the metric proposed in the paper). We also measured the average number of sections that an entity is mentioned in, but we excluded this metric from the final results, as we found that this finer-grained metric gave similar results to our easier to calculate metric presented above.

    b. **Entity consistency**: We consider as attributes of an entity all verbs and adjectives, since these parts of speech are commonly used to describe an entity. We avoid using nouns as well, since that would confuse attributes with entities and it would be difficult to accurately discriminate between the two. At the example of Figure 1b the inconsistency mostly lies on the use of nouns, but if you consider all these different nouns as different identified entities (mother, sister, niece), then you have a lot of different entities associated with the same verb ‘am’, which would result in a low consistency score.
For simplicity, we consider entity attributes in a sentence-level basis. When multiple entities are mentioned in the same sentence, we consider all attributes within the sentence associated with all corresponding entities. This is a limitation of our current metric, which could be further refined by considering a dependency parser to more accurately assign attributes to entities. However, since a dependency parser would make the analysis of the original and generated stories slower and would introduce extra errors, we leave this direction for future work.
Finally, regarding the low correlation between the automatic metric and the human ratings for entity consistency, we have made two main observations. First, although the correlation here is low, when we compare the consistency between human-written and generated stories in Table 2, we observe that there is a large gap for both datasets. This indicates that this metric is able to capture in some degree the shortcomings of LMs and there is still plenty of room for improvement. Second, in this work we conclude that judging consistency is generally challenging, even for human annotators, since consistency has also the lower correlation with overall preference in the human evaluation (Table 5). Automatically measuring consistency is also challenging, as it is potentially confounded by entity coherence; we discuss this more in the next section.

---

> ### Author Response · Authors · 2021-11-16
> **General Response (continued)**
>
> 2. **Automatic Results for WikiPlots**
> One limitation of the automatic computation of consistency is that it does not take into consideration different levels of entity coherence. However, in practice consistency is dependent on coherence. For example, if a LM only mentions an entity once throughout a story, consistency is guaranteed because there are no other mentions to contradict previous statements. Hence, higher entity coherence (i.e., more mentions of an entity in longer spans of text) in general results in lower consistency scores, all else being equal. This is the reason why automatically comparing the two models in terms of consistency in the WikiPlots dataset is not straightforward. Our model mentions each entity twice as often as the VanillaLM on average and presents 57% relative improvement in terms of coherence, which makes the consistent assignment of attributes to entities more challenging. However, the two models are comparable in terms of coherence in the WritingPrompts dataset, where we can more accurately compare consistency and we observe an advantage when using the dynamic entity memory. Moreover, we use the same stories from the WikiPlots dataset for the human evaluation experiment, and validate that according to human judges we are able to maintain the same degree of consistency while we significantly improve coherence in the generated stories. Our conclusion here is that we should first achieve a desired degree of entity coherence, before being able to accurately investigate consistency.
>
> 3. **Human Evaluation**
> Regarding the human evaluation criteria and the corresponding scaling, see Figure 5 of the Appendix, where we provide a detailed example of the template used during human evaluation. Finally, we asked 2 different human judges to evaluate each story, and we have updated the paper to include inter-annotator agreement metrics (see Table 6 of the Appendix).

---

### Author Response · Authors · 2021-11-16
**Summary of Paper Revisions**

We would like to thank the reviewers for helping us improve the clarity of our paper. We updated the paper as follows:

1. We changed Figure 1(a) based on the suggestion of Reviewer 2yed.

2. We improved the discussion regarding model performance on the WikiPlots dataset according to the automatic metrics (Section 5.2).

3. We improved the discussion regarding the low correlation between the automatic consistency metric and human ratings in Section 5.3.

4. We added the proposed reference for plot-based coherence in story generation in Section 6 based on the comment of Reviewer XC2i.

5. We added details about the human evaluation setup (Section A.3 of the Appendix and Table 6).

6. We added the appropriate pointers in the paper for the details provided in the Appendix (especially for human evaluation and examples of generated text with marked entity patterns).

---

### Decision · Program_Chairs · 2022-01-20

**Decision:**

Reject

**Comment:**

This work analyzes the ability of pre-trained language models to maintain entity coherence and consistency in long narrative generation. Along with new automatic metrics for analyzing narrative generation, it proposes a memory-augmented model that allows tracking entities to improve narrative generation.  Although all the reviewers appreciated the importance of the problem, the novelty of the proposed approach, as well as empirical improvements in a subset of experiments, they also acknowledge several major weaknesses including the lack of rigor in defining the method, the lack of clarity in writing (especially in the experiments section), insufficiently strong baselines, and an issue of reproducibility since the code cannot be released. These concerns were in part addressed during rebuttal, but not enough to accept the paper.